# Mechanism of substrate hydrolysis by the human nucleotide pool sanitiser DNPH1

Neil J. Rzechorzek [1], Simone Kunzelmann [2], Andrew G. Purkiss [2], Mariana Silva Dos Santos [3], James I. MacRae [3], Ian A. Taylor [4], Kasper Fugger[1,5] & Stephen C. West [1] ✉

Poly(ADP-ribose) polymerase (PARP) inhibitors are used in the clinic to treat *BRCA*-deficient breast, ovarian and prostate cancers. As their efficacy is potentiated by loss of the nucleotide salvage factor DNPH1 there is considerable interest in the development of highly specific small molecule DNPH1 inhibitors. Here, we present X-ray crystal structures of dimeric DNPH1 bound to its substrate hydroxymethyl deoxyuridine monophosphate (hmdUMP). Direct interaction with the hydroxymethyl group is important for substrate positioning, while conserved residues surrounding the base facilitate target discrimination. Glycosidic bond cleavage is driven by a conserved catalytic triad and proceeds via a two-step mechanism involving formation and subsequent disruption of a covalent glycosyl-enzyme intermediate. Mutation of a previously uncharacterised yet conserved glutamate traps the intermediate in the active site, demonstrating its role in the hydrolytic step. These observations define the enzyme's catalytic site and mechanism of hydrolysis, and provide important insights for inhibitor discovery.

Individuals with inheritable mutations in the *BRCA1* or *BRCA2* tumour suppressor genes are predisposed to breast, ovarian and prostate cancers[1]. In the clinic, these patients are treated with inhibitors of poly [ADP-ribose] polymerase (PARPi), which cause PARP trapping (increased residence time) at single-strand breaks in DNA, leading to replication fork collapse and cell death[2–5]. While effective at initial cancer maintenance, after a period of time the tumours develop resistance to PARP inhibition leading to further growth[6,7]. Recently, we discovered that loss or inhibition of a nucleotide pool sanitiser 2′-deoxynucleoside 5′-monophosphate N-glycosidase (DNPH1) potentiates the sensitivity of *BRCA*-deficient cells to PARPi, offering a promising strategy for improved therapy for these individuals[8].

The cellular role of DNPH1 is to remove hydroxymethyl deoxyuridine monophosphate (hmdUMP) from the nucleotide pool. This activity prevents a cascade of nucleotide phosphorylation events that involves deoxythymidylate kinase (DTYMK), thereby limiting the incorporation of hmdU into genomic DNA[8]. hmdUMP is a metabolic product that arises from the epigenetically modified nucleotide hydroxymethyl deoxycytidine, and the products of hmdUMP hydrolysis by DNPH1 are hydroxymethyl uracil (hmU) and deoxyribose 5′-phosphate (dRP)[8] (Fig. 1). In the absence of DNPH1, hmdU incorporation leads to aberrant base removal by the SMUG1 glycosylase, strand incision by apurinic/apyrimidinic (AP) endonuclease-1, and PARP trapping. As a consequence, replication forks collapse leading to DNA double strand breaks (DSBs) and apoptosis.

DNPH1 belongs to a family of retaining N-glycosidases that utilise a double displacement mechanism in which a glycosyl-enzyme intermediate is formed and hydrolysed by acid/base catalysis, mediated by the carboxylic side chains of aspartic or glutamic acid[9,10] (Supplementary Fig. 1). In the first step, protonation by a catalytic aspartate (D80) drives nucleobase removal, resulting in an ester linkage between the remaining sugar-phosphate moiety and a catalytic glutamate

[1]DNA Recombination and Repair Laboratory, The Francis Crick Institute, 1 Midland Road, London NW1 1AT, UK. [2]Structural Biology Science Technology Platform, The Francis Crick Institute, 1 Midland Road, London NW1 1AT, UK. [3]Metabolomics Science Technology Platform, The Francis Crick Institute, 1 Midland Road, London NW1 1AT, UK. [4]Macromolecular Structure Laboratory, The Francis Crick Institute, 1 Midland Road, London NW1 1AT, UK. [5]Present address: University College London Cancer Institute, 72 Huntley Street, London WC1E 6DD, UK. ✉e-mail: stephen.west@crick.ac.uk

**Fig. 1 | DNPH1 cleaves the glycosidic bond in hmdUMP.** Hydrolysis generates deoxyribose monophosphate (dRP) and hydroxymethyl uracil (hmU) products. Phosphate, deoxyribose and hmU moieties are coloured orange, black and green respectively.

(E104)[11]. In the second step, the aspartate activates an incoming water molecule to cleave the nascent glycosyl ester, and the sugar-phosphate is released.

Structural analyses of rat and human DNPH1 (also known as RCL) revealed a homodimeric arrangement of DNPH1 in ligand-bound and ligand-free states[12–17]. However, when most of these studies were carried out the cellular substrate of DNPH1 was unknown, and the structural characterisations made use of either inhibitory ribose-type, purine-based compounds or non-canonical ligands that fail to mimic the natural hmdUMP substrate. Here, we present crystal structures of dimeric human DNPH1, bound to hmdUMP, at different stages of the catalytic cycle. In the first, an inactivating E104Q mutation permits binding of the intact substrate without cleavage. A conserved helical element guarding the entrance to the active site presents a histidine (H56) for interaction with the base, accelerating substrate cleavage by the active enzyme. In the second structure, mutation of the glutamate (E55) adjacent to H56 results in trapping of a glycosyl-enzyme intermediate, indicating an important role for this residue in the second hydrolytic step.

## Results

### Crystal structure of DNPH1 bound to hmdUMP

To understand the mechanism of substrate recognition by DNPH1, a crystallisation construct encoding amino acids 19–162 with an inactivating E104Q point mutation[8] was designed based on existing literature[12–16]. The purified protein (referred to as DNPH1[E104Q]) was co-crystallised with hmdUMP at 4 °C and X-ray diffraction data were collected from a single crystal. The structure was solved in space group P2$_1$ at a resolution of 1.78 Å by molecular replacement using PDB code 4P5E [https://doi.org/10.2210/pdb4p5e/pdb] as a search model[15]. Supplementary Table 1 provides final data and model quality statistics.

The six protein chains found in the asymmetric unit present as two full biologically active homodimers (Fig. 2a) and two half homodimers; corresponding partners to the latter are defined by crystallographic symmetry. All copies are bound to substrate and are remarkably similar to each other (RMSD between Cα atom pairs = 0.128–0.406 Å), although there is some disorder for amino acids ~60–70, which form a loop across the top of the active site binding pocket. While only two of these 'shielding' loops are fully visualised, they have not previously been resolved in ligand-bound structures for DNPH1. Apart from these loops, each substrate-bound subunit is similar to the inhibitor-bound search model used for molecular replacement (Supplementary Fig. 2a).

As described previously[13], each DNPH1 monomer adopts a Rossmann-like fold consisting of five core β-strands surrounded by five major α-helices (Supplementary Fig. 2b). Two smaller α-helices are noteworthy: an additional helical element comprising amino acids 54–59, and a helical 'extension' comprising amino acids 128–133. Three key regions of each subunit show high levels of conservation

(Supplementary Fig. 2c): (i) the active site, formed by the second and third α-helices and the first β-strand, and abutted by the additional helical element, (ii) the dimerisation interface generated by the third α-helix, and (iii) the helical extension that makes additional contacts with the active site of the opposing chain. The shielding loop, which precedes the second α-helix, is not conserved. The core catalytic triad, consisting of Y24, D80 and E104Q, sits below the plane of the substrate deoxyribose sugar, poised for nucleophilic attack at the anomeric carbon (Fig. 2b).

The base, which is held in plane by hydrophobic interactions from I29 and I76 side chains (Supplementary Fig. 2d), adopts an *anti* conformation, with chi torsion angles (defined by O4′-C1′-N1-C2) of −178.9°, or 174.5 to 179.6°. The phosphate group is strongly coordinated by S98 and S128′, the latter being donated by the helical extension (amino acids 128′–133′) from the adjacent subunit (Fig. 3a). The same extension additionally positions the substrate hydroxymethyl group through a backbone amide interaction. This enables the substrate to stabilise its own phosphate, with the hydroxymethyl mimicking the coordination provided by the two serine residues. The combination of intra- and inter-molecular interactions drives, at least in part, the substrate specificity of DNPH1. At the other end of the substrate, the inactivating E104Q mutation introduces an amine group that forms an additional hydrogen bond with D80, pulling both catalytic residues away from the sugar (Fig. 3b). An additional interaction is observed between a base carbonyl oxygen (O2) and the sidechain of H56, which sits on the helical element immediately upstream of the shielding loop. Where visualised, the shielding loop makes no obvious contacts directly with the substrate and appears to be highly mobile.

### Targeting hydroxymethyl pyrimidines—conservation of substrate binding

The mode of substrate binding by DNPH1 is reminiscent of the enzyme MilB, a bacterial N-glycosidase that preferentially hydrolyses 5-hydroxymethyl-cytidine 5′-monophosphate (hmCMP)[18]. MilB is also dimeric (PDB 4OHB [https://doi.org/10.2210/pdb4ohb/pdb]; Supplementary Fig. 3a) and each individual substrate-bound subunit bears remarkable similarity to those of DNPH1[E104Q] (Fig. 3c). Subtle differences between the respective substrates of the two enzymes are reflected in key residue variations. To accommodate a ribose rather than deoxyribose substrate, the MilB equivalent to DNPH1's Y24 is F17 (Fig. 3d), consistent with observations showing that a F17Y mutation reverts substrate preference toward deoxyribose moieties[19]. Additionally, in order to bind the two accessible amine groups on the cytidine base, MilB utilises a glutamate (E62) in place of the equivalent H56 in DNPH1 (Fig. 3d).

MilB and DNPH1 differ in the way that they coordinate the phosphate. While MilB retains the equivalent to S98 in DNPH1 (S97 in MilB), the incoming helical extension is missing a coordinating serine, and sits too far away to stabilise the substrate hydroxymethyl group

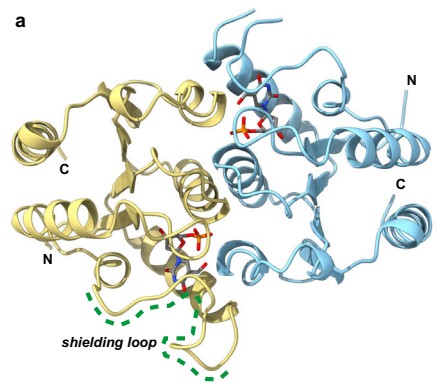

**Fig. 2 | X-ray crystallography illuminates substrate recognition by DNPH1.**
**a** Cartoon representation of dimeric DNPH1[E104Q] (chain B in yellow, chain A in light blue) bound to hmdUMP substrate (shown in sticks). A shielding loop (amino acids ~60–70, green dotted lines) is fully visualised in chain B. N- and C-termini are labelled. **b** Close-up view of hmdUMP bound to the active site of DNPH1[E104Q] chain B (yellow), supplemented by interactions from chain A (light blue). The substrate and sidechains of interacting residues are shown as sticks. The electron density for the bound ligand contoured to 2 σ is shown as a green mesh. Ligand/sidechain oxygen, nitrogen and phosphorus atoms are coloured red, blue and orange, respectively.

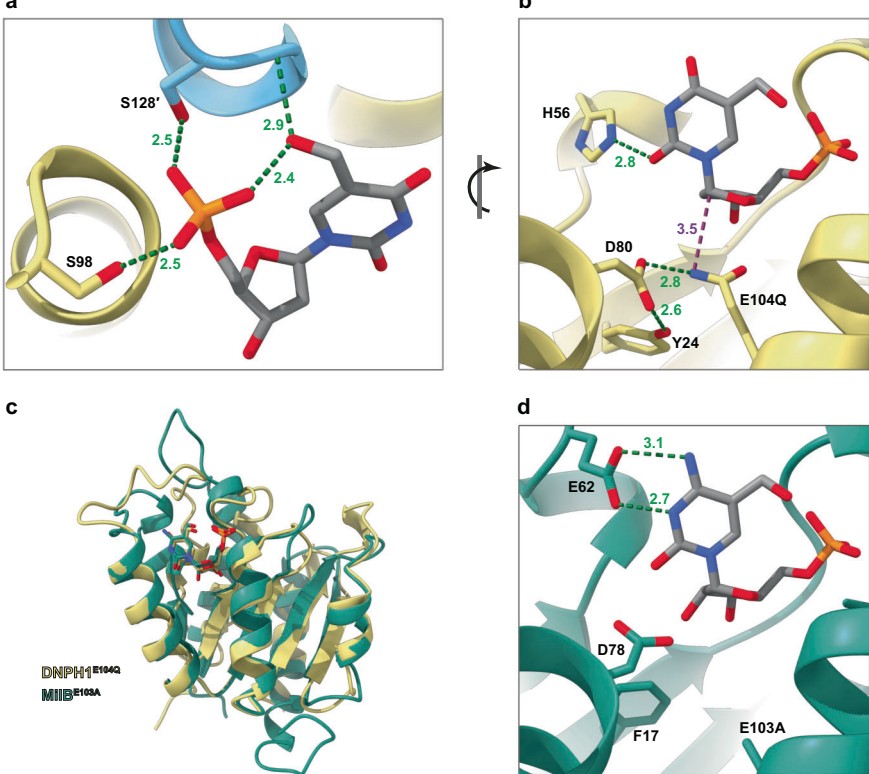

**Fig. 3 | Structural studies reveal conserved functional residues in DNPH1.**
**a**, **b** Focused views of hmdUMP bound to the active site of DNPH1[E104Q] chain B (yellow) supplemented by interactions from chain A (light blue). Green dotted lines highlight key hydrogen bond interactions. **a** Intra- and inter-molecular interactions stabilise ligand phosphate and hydroxymethyl moieties. **b** Rotated view of (**a**) highlighting active site arrangements and substrate positioning. E104Q is inappropriately oriented for nucleophilic attack (magenta dotted line) at the anomeric carbon. H56 interacts with the base carbonyl oxygen O2. **c** Superposition of DNPH1[E104Q] chain B (yellow) bound to hmdUMP with MilB[E103A] (PDB 4OHB; green) bound to hmCMP. RMSD between 77 Cα atom pairs = 0.902 Å. The relative positions of ligands are remarkably similar. **d** Close-up view of hmCMP (grey sticks) bound to the active site of MilB[E103A] (green), for comparison with DNPH1[E104Q] binding to hmdUMP, as shown in **b** Fig. 2b E62 stabilises the substrate via hydrogen bonds (green dotted lines) with nitrogen moieties on the base. Sidechains of active site/interacting residues are shown as sticks; ligand/sidechain oxygen, nitrogen and phosphorus atoms are coloured red, blue and orange respectively. Bond distances are given in Å to 1 decimal place.

(Supplementary Fig. 3b). Instead, MilB makes use of a conserved arginine (R23) to satisfy these two key interactions[18]. Although this residue is retained in DNPH1 (R30), it is instead involved in base recognition. Stabilised by a salt bridge with D73, R30 uses a water molecule to interact with the remaining base carbonyl oxygen O4 (Supplementary Fig. 3c). DNPH1 therefore uses a combination of

secondary structure elements and conserved residues to recognise all oxygen-bearing groups on the substrate base.

Multiple sequence alignment of fully annotated DNPH1 protein sequences (Supplementary Fig. 3d) shows that the key base recognition residues R30 and H56 are conserved. Mutation of these residues (DNPH1[R30A], DNPH1[H56A]) resulted in a ~6-fold and ~11-fold reduction in

substrate turnover relative to wild-type enzyme (Fig. 4a), respectively, in keeping with structural observations. When catalytic rates at different substrate concentrations were measured using a continuous spectrophotometric method that observed shifts in the UV-absorbance maximum following hmU liberation, we found that mutation of H56 resulted in only a modest increase in the enzyme's Michaelis constant, $K_m$ (Fig. 4b, c). These results lead us to suggest that loss of substrate affinity does not fully account for the observed reduction in catalytic rate.

## A conserved histidine accelerates base cleavage

To determine the function of H56 in substrate turnover, we developed a simplified mechanistic model describing the key rate-limiting steps of catalysis (see Methods). Briefly, this involves: the rate constants of association ($k_1$) and dissociation ($k_{-1}$) between enzyme (E) and substrate (S) during formation of the Michaelis complex (ES); the first catalytic rate constant ($k_2$) describing the formation of the glycosyl-enzyme intermediate (EI); and the second catalytic rate constant ($k_3$) describing cleavage of the sugar-phosphate from the enzyme.

Relying on the intrinsic fluorescence of a tryptophan residue (W83) adjacent to the active site, we utilised stopped-flow techniques to determine which catalytic rate constant ($k_2$ or $k_3$) is rate-limiting for various DNPH1 point mutants. Measurements made at short timescales with a single substrate concentration (Fig. 4d) revealed that the binding and/or hydrolysis of hmdUMP elicited different fluorescence signatures across the various mutants tested.

The wild-type enzyme, DNPH1$^{WT}$, presented a biphasic response; a rapid reduction in fluorescence was followed by a partial recovery and a temporary plateau. At longer timescales (Supplementary Fig. 4a) additional signal recovery was retarded at higher substrate concentrations, indicative of substrate depletion. Shorter timescales showed that the magnitude and rate of initial signal loss correlated with increasing substrate levels (Supplementary Fig. 4b), indicating that this first phase represents substrate binding.

In contrast, the long and short timescale data for the catalytically inactive mutant DNPH1$^{E104Q}$ presented as a single phase, consisting of a rapid decrease in fluorescence intensity that was dependent on substrate concentration (Supplementary Fig. 4c and d). In the absence of substrate turnover, this signature is indicative of the rapid build-up of the Michaelis complex, ES. We therefore conclude that the rapid partial signal recovery seen with wild-type DNPH1 indicates a reduction in ES complex levels and subsequent build-up of the glycosyl-enzyme

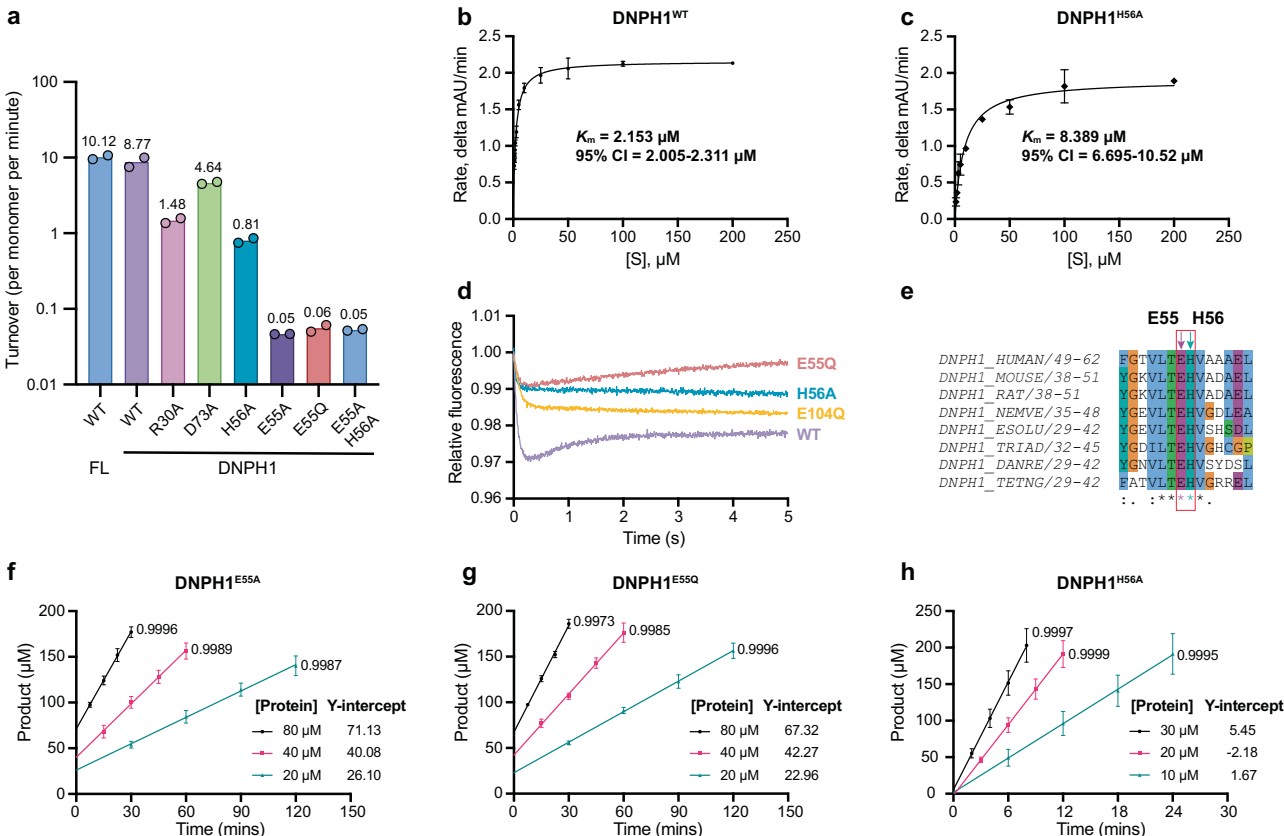

**Fig. 4 | Activity assays reveal key residues required for substrate hydrolysis. a** Substrate turnover rates for various DNPH1 constructs used in this study, as measured by HPLC. Full-length and wild-type constructs are labelled FL and WT, respectively. Crystallisation constructs (aa 19-162) are indicated by the suffix DNPH1. Bars and values indicate the mean from $n = 2$ independent experiments. Filled circles represent individual data points. **b**, **c** Turnover rate versus [hmdUMP] curves for DNPH1$^{WT}$ and DNPH1$^{H56A}$. Data points and error bars represent the mean and standard deviation, respectively, from $n = 3$ independent experiments. Curves are the standard Michaelis–Menten model fitted to the data by non-linear regression, yielding the Michaelis constant ($K_m$) for each enzyme. **d** Normalised short timescale stopped-flow measurements of tryptophan fluorescence during substrate binding/hydrolysis by the DNPH1 mutants. Representative plots of single datasets are averages of three technical repeats for 2 μM enzyme and 50 μM substrate; each experiment was independently repeated three times. **e** Multiple sequence alignment of a conserved LTEHV motif in DNPH1 protein sequences. Symbols underneath each row indicate consensus level: '*', a single, fully conserved residue, ':', conservation of residues with strongly similar properties, and '.', conservation of residues with weakly similar properties. Ligand-binding residue H56 and adjacent E55 are indicated with teal and magenta arrows respectively. **f–h**, Time course plots for the indicated DNPH1 mutants in substrate turnover experiments, as measured by HPLC. Data points and error bars represent the mean and standard deviation, respectively, from $n = 2$ independent experiments. Goodness of fit (R-squared) is indicated for each line of best fit. Source data are provided as a Source Data file.

intermediate (EI) during substrate turnover. These results show that the second catalytic step ($k_3$), and not the first ($k_2$), is rate-limiting for the wild-type enzyme in the steady-state.

The long and short timescale data for DNPH1$^{H56A}$ resembled those of DNPH1$^{E104Q}$, and presented as a single phase that indicates maintained build-up of the ES complex (Supplementary Fig. 4e and f). Since DNPH1$^{H56A}$ retains activity, albeit ~11-fold reduced in comparison with DNPH1$^{WT}$, we infer that the first catalytic step ($k_2$), rather than the second ($k_3$), is rate-limiting. These results demonstrate that H56 plays a direct role in catalysis, possibly by stabilising the hmU leaving group during glycosidic bond cleavage. This stabilisation accelerates the first catalytic step, as observed with other DNA repair glycosylases.

### A conserved glutamate releases the glycosyl-enzyme intermediate

Close inspection of the multiple sequence alignment revealed that H56 resides on a conserved motif (aa. 53–57, LTEHV) (Fig. 4e). This motif broadly overlaps with the additional helical element (aa. 54–59) and includes an adjacent glutamate at amino acid 55 that appears to be invariant across metazoan DNPH1 sequences[20]. However, the glutamate is largely solvent exposed and makes no obvious interactions in our ligand-bound structure, giving no indication of the reason for its conservation. Given the importance of a similarly positioned glutamate for substrate recognition in MilB, we investigated the role of E55 in DNPH1 activity. Surprisingly, mutation of this residue, either to alanine (E55A) or glutamine (E55Q), resulted in a ~150-fold reduction in substrate turnover (Fig. 4a).

Analysis of reaction progress curves (Fig. 4f, g) indicated DNPH1$^{E55A}$ and DNPH1$^{E55Q}$ displayed rapid burst kinetics; the burst amplitude (the extrapolation of the amount of hmU product at time $t = 0$) was approximately equal to the concentration of enzyme in the system. These results indicate that both mutants rapidly cleave a single substrate molecule before reducing to a much slower rate of catalysis. Therefore, E55 appears to be involved in the second step of the catalytic cycle in which the glycosyl-enzyme intermediate is released. Conversely, DNPH1$^{H56A}$ did not display this phenomenon (Fig. 4h). We observed that the activity of the DNPH1$^{E55A\ H56A}$ double mutant was similar to that of the single E55A or E55Q mutants (Fig. 4a). Collectively, these data indicate that each respective residue plays a key role in separate stages of the catalytic cycle, only one of which is rate-limiting in any given point mutant.

Both long and short timescale stopped-flow data for DNPH1$^{E55Q}$ (Supplementary Fig. 4g, h) resembled those of DNPH1$^{WT}$, in which a rapid decrease in fluorescence intensity was followed by a gradual recovery. As for the WT enzyme, this indicates that the second catalytic step ($k_3$) is rate-limiting, in agreement with the reaction progress curve data. Analysis of all stopped-flow traces for each point mutant yielded observed rate constants for the phases presented (Supplementary Fig. 4i, j). These data provide approximations for substrate affinities ($K_D$) in all cases, and estimation of the non-limiting rate constant $k_2$ for DNPH1$^{WT}$ and DNPH1$^{E55Q}$ (Supplementary Table 2). Steady-state substrate turnover data complete the picture by providing estimates for the rate-limiting constants for each mutant.

From these analyses, we determined that the first catalytic step in DNPH1$^{WT}$ is approximately 8-fold faster than the second. However, the H56 mutation resulted in a ~120-fold rate reduction in the first catalytic step, demonstrating the importance of this residue in glycosidic bond cleavage. Meanwhile, mutation of E55 resulted in a ~150-fold rate reduction in the second catalytic step, and only a marginal (~6-fold) reduction in the first, confirming the role of this residue in cleavage of the glycosyl-enzyme intermediate. In both mutants, substrate affinity was reduced only 2-fold.

### Snapshot of a catalytic intermediate

To visualise the trapping of the glycosyl-enzyme intermediate, we employed a crystallisation construct harbouring a single E55Q point mutation. Purified DNPH1$^{E55Q}$ protein was co-crystallised with hmdUMP and the structure solved in space group P2$_1$2$_1$2$_1$ at 1.65 Å resolution using the DNPH1$^{E104Q}$ structure for molecular replacement.

In the crystal structure a biologically active homodimer comprises the asymmetric unit, and both active sites contain a trapped glycosyl-enzyme intermediate with deoxyribose monophosphate (dRP) covalently fused to E104. However, in both chains, the dRP:E104 moiety adopts two distinct, alternative conformations, one of which permits an additional interaction between the constituent sugar and surrounding sidechains. The base of the substrate has been cleaved and is no longer visible, but has been largely replaced by a well-defined water network (Fig. 5a, b). This network is stabilised in part by E55Q, which is now presented into the active site through reorganisation of the helical element harbouring the LTEHV motif. Consequently, H56 tucks away below the anomeric carbon. For both chains, the shielding loop (residues ~60–70) is disordered.

Remarkably, E55Q appears to be ideally positioned although, due to the mutation, chemically unable to activate one of the bound waters for nucleophilic attack at the anomeric carbon, in agreement with activity assays. Crucially, the conserved catalytic aspartate (D80) is too far away to perform this activation, and the active site arrangement indicates that this is also the case in the wild-type enzyme. This is at odds with the broadly accepted mechanism of catalysis for this family of hydrolases, which classically rely on the same two carboxylate groups for both stages of the catalytic cycle. As such, we believe we have identified E55 as an additional contributor at the DNPH1 active site that alters our understanding of this enzyme's catalytic mechanism.

In both chains, H56 adopts two rotameric conformations that correlate with the two alternative conformations of the dRP:E104 moiety. To avoid local clashes, these two conformations are also linked to smaller movements of residues Y24 and C26. Accordingly, we constrained conformational occupancies for these species in each separate chain during refinement. In chain A (Supplementary Fig. 5a–c), the two alternative conformations refine to approximately 50% occupancy each. This results in poor sidechain density for each H56 rotamer, but yields good evidence for both dRP:E104 conformations. However, in chain B (Supplementary Fig. 5d–f), conformation 'A' refines to nearly 80% occupancy, enabling clearer definition of the interaction network. In this dominant arrangement, H56 electrostatically drives the dRP 3' hydroxyl group to form a hydrogen bond with the remaining sidechain carbonyl oxygen of E104 (Fig. 5c). In contrast, in conformation 'B', which is best visualised in chain A (Fig. 5d), this additional dRP-E104 hydrogen bond is not possible.

## Discussion

Human nucleotide glycosidase DNPH1 represents an important drug target that, in combination with PARPi's, offers a new avenue for treatment of *BRCA*-deficient cancers. In this study, we have presented crystallographic structures of the enzyme bound to its substrate at two key stages of the catalytic cycle. In combination with substrate turnover assays, these structures reveal important mechanistic insights that provide two alternative starting points for rational drug design.

In the first structure, an inactivating E104Q point mutation permits the binding of hmdUMP without subsequent cleavage. The mode of binding is similar, though subtly different, to that of the bacterial enzyme MilB, which targets the closely related nucleotide hmCMP. For example, both enzymes use functionally conserved interactions to generate intramolecular contacts between phosphate and hydroxymethyl moieties on their respective substrates. It is plausible that these increase glycosidic bond strain, reducing the energy barrier for hmU liberation.

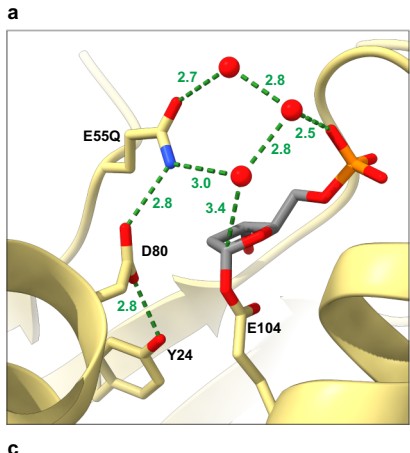

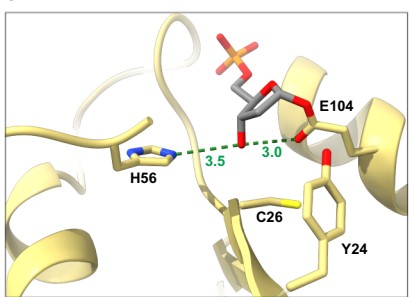

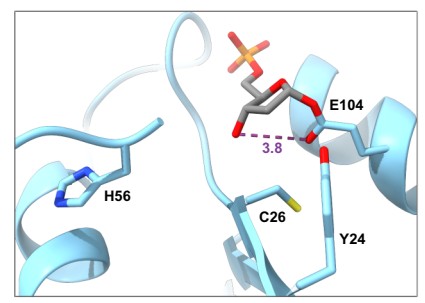

**Fig. 5 | DNPH1^{E55Q} traps a glycosyl-enzyme intermediate. a** Close-up view of deoxyribose monophosphate covalently bound at the active site of DNPH1^{E55Q} following initial cleavage of hmdUMP. A stabilised water network replaces the missing base. E55Q maintains the network and appears ideally positioned for water activation, though the amide group disables this. D80 remains held away from the active site. Model shown for chain B (yellow, conformation A). **b** as (**a**), with electron density map (green mesh) contoured to 2 σ around bound ligand, water network and E55Q/E104 sidechains. **c, d** Two H56 rotamers subtly alter dRP positioning and nearby residues (single conformations shown). In (**c**), chain B (yellow), H56 predominantly adopts conformation A, driving the dRP 3′ hydroxyl group to interact with E104, weakening the glycosidic bond. In (**d**), chain A (blue) in conformation B, H56 sits away from the ligand, reducing the secondary dRP:E104 interaction. Ligand/sidechain oxygen, nitrogen and phosphorus atoms are coloured red, blue and orange, respectively. Likely/unlikely hydrogen bonds shown as green/purple dotted lines, respectively. Bond distances given in Å to 1 decimal place.

In DNPH1, the enzyme contacts all oxygen-bearing groups of the substrate base. Notably, the interaction of H56 appears to play a role in catalysis beyond that of substrate binding. One possibility is that this residue stabilises the leaving group as a negative charge develops, lowering the energy barrier for glycosidic bond cleavage and thereby accelerating the reaction. This is reminiscent of the uracil DNA glycosylase (UDG) superfamily of enzymes, in which many members rely on a similarly positioned histidine to remove uracil-type bases from DNA during base excision repair. While these enzymes differ catalytically, as base flipping rather than acid catalysis is used to drive cleavage by applying glycosidic bond strain, the histidine enables the leaving group to depart as an anion[10,21]. Of note, one member of this family of glycosylases, SMUG1, can excise hmU from DNA[22], indicating that the additional hydroxymethyl group does not render the overall catalytic approach unfeasible. In the absence of base flipping, which can only take place in double-stranded DNA, it is tempting to speculate that DNPH1 combines acid hydrolysis via D80 with leaving group stabilisation via H56 to facilitate glycosidic bond cleavage. Further studies will be required to determine the precise contribution made by each residue.

In the second structure, the E55Q point mutation traps DNPH1 part way through its catalytic cycle, with a covalent dRP:E104 moiety present in the active site. The complete rearrangement of the additional helical element removes H56 from the base-binding pocket, and positions E55Q to facilitate water activation and cleavage of the nascent ester linkage. While this is at odds with the broadly accepted mechanism, in which D80 is expected to act as the general base, the structure presented here indicates that the catalytic aspartate would be poorly positioned for water deprotonation. Although it could be

argued that the sidechain amine on E55Q might artificially restrict D80 from correct placement, no such restriction would exist for an E55A point mutant (substrate turnover assays for DNPH1^{E55Q} and DNPH1^{E55A} reveal similar burst phenomena and steady-state turnover rates). Therefore, we suggest that DNPH1 is different from other retaining glycosidases, in that it uses separate residues for their respective acid/base properties in the two halves of the catalytic cycle.

The DNPH1^{E55Q} structure also reveals that H56 might perform a supporting role in dRP release, as it appears to drive the deoxyribose to form an additional hydrogen bond with E104. This secondary interaction of a nucleophilic carboxylate with a sugar hydroxyl group has been seen previously in the covalently trapped structure of the β-1,4-glycanase, CEX[23]. We speculate that in DNPH1, this secondary interaction assists in destabilising the nascent dRP:E104 ester linkage, possibly through conformational strain in the sugar ring, to accelerate dRP removal and enzyme reset. However, the effects of this interaction would appear to be small, given that turnover assays indicate a more prominent role for H56 in substrate recognition and/or leaving group stabilisation.

Finally, we note that DNPH1 is closely related to the nucleoside 2′-deoxyribosyltransferase (NTD) family of enzymes[12], which catalyse the transfer of deoxyribose between a nucleoside and an acceptor nucleobase. While there are clear differences (for example, NTD enzymes lack the conserved phosphate binding pocket found in DNPH1), the similarities in overall fold and catalytic function are instructive. NTD enzymes fall into two main categories[24]: (i) class I enzymes, which exclusively catalyse transfer between purine bases, and (ii) class II enzymes, which catalyse transfer between both purine and pyrimidine bases. In the latter case, binding of a pyrimidine

nucleoside is accompanied by the positioning of a loop across the top of the base, protecting the active site from solvent exposure[25]. This loop appears to be similarly positioned to the DNPH1 shielding loop, implicating both structures in enzyme function.

Taken together, our structural and biochemical data enable us to propose a model for the hydrolysis of hmdUMP by DNPH1 (Fig. 6). Substrate binding is driven by phosphate coordination and both intra- and inter-molecular interactions that target all oxygen-bearing groups of the departing nucleobase. H56 positions and likely stabilises the hmU leaving group, which is ultimately protonated by D80 upon glycosidic bond cleavage. Nucleophilic attack by E104 enables cleavage to proceed and results in the formation of a covalent glycosyl-enzyme intermediate. During this first part of the catalytic cycle, a shielding loop (aa. ~60–70) guards the active site and likely hinders unproductive substrate release. Departure of the cleaved nucleobase requires the shielding loop to move, enabling reorganisation of the additional helical element (aa. 54–59). The latter structural change positions E55 for water activation, driving cleavage of the nascent glycosyl-ester bond. H56 is repositioned to aid this second step by enforcing an interaction between E104 and the ribose hydroxyl, and we suggest that this may stabilise developing charge and generate conformational strain in the sugar. The enzyme resets when the cleaved dRP departs, enabling fresh substrate binding.

This model has important implications for novel DNPH1 inhibitor design. Substrate mimics, or perhaps transition-state mimics that follow the principles of the immucillins[26], will likely take advantage of the interactions between the nucleobase and conserved residues such as

H56 and R30. Notably, the substrate base is surrounded by a water network that is encapsulated by the shielding loop (Supplementary Fig. 6a). This network can act as a marker for space into which novel compounds can be built during rational drug design. Indeed, known purine-based DNPH1 inhibitors (Supplementary Fig. 6b) already exploit this region to a certain extent. However, targeting the second phase of the catalytic cycle may also be fruitful. Substitutions to the sugar, as found in fluorine derivatives, have been widely used to trap glycosyl-enzyme intermediates, and could be of utility against DNPH1, especially given the role of H56 in ribosyl hydroxyl positioning. Furthermore, any substitution that prevents proper access of E55 into the active site, or which prevents correct positioning of the activated water, will be worthy of investigation.

## Methods
### Cloning
A full-length construct encoding isoform 1 of human DNPH1[8] was silently mutated using QuikChange II (Agilent Technologies) to remove an encoded *Nco*I restriction site. The gene was subsequently amplified by PCR and inserted into the pOP1b subcloning vector (a gift from M. Hyvönen, unpublished) using *Nco*I and *Xho*I restriction sites. The resulting pOP1b-DNPH1 construct was then used for further site-directed mutagenesis rounds, where necessary.

To produce truncated proteins suitable for structural studies, genes encoding residues 19–162 were amplified by PCR from full length constructs using oligonucleotides *FwdNcoI-19* (5′-CTAGC-CATGGGCCGCCCGGCCCTGTA-3′) and *RevXhoI-162* (5′-CTAGCTCGA

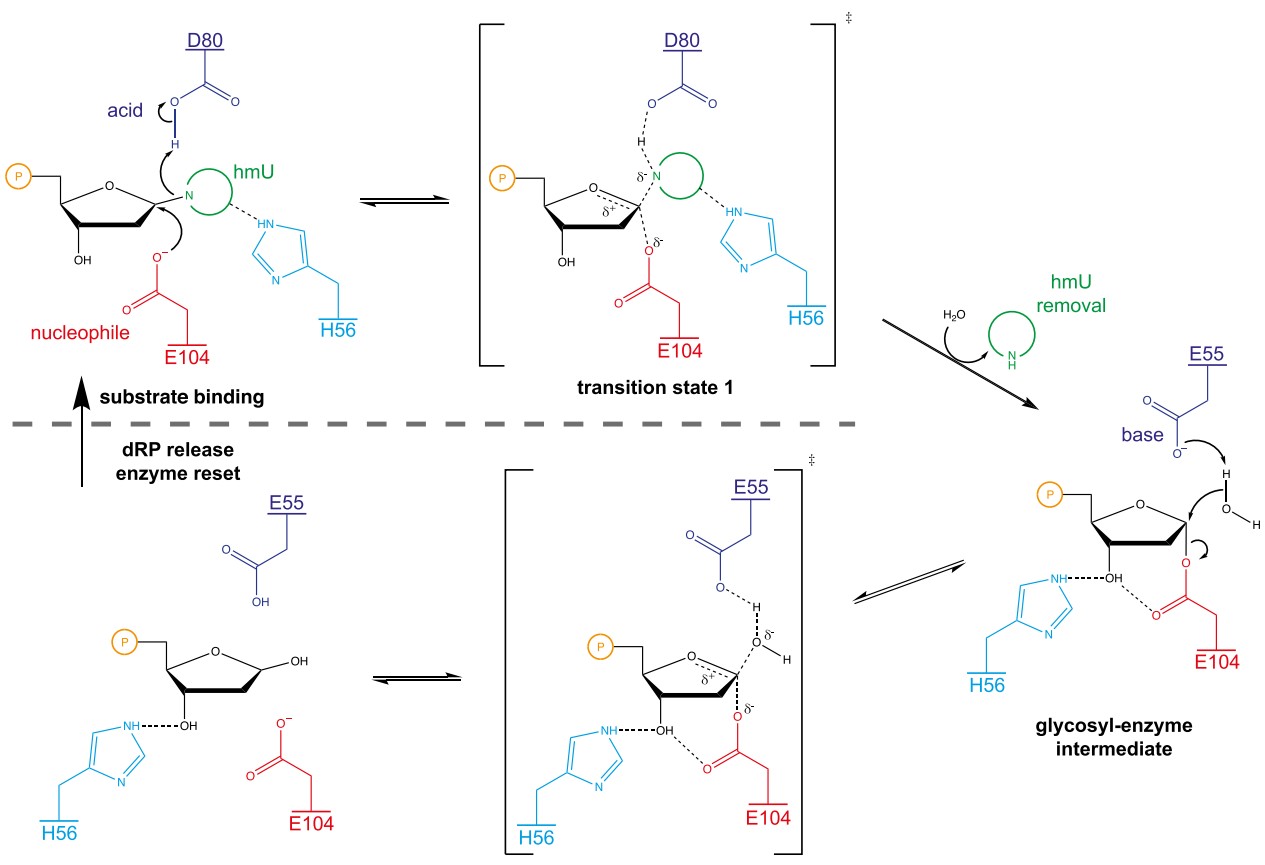

**Fig. 6 | Mechanism of cleavage of hmdUMP by DNPH1.** H56 stabilises the hmU leaving group, which is protonated by D80 upon glycosidic bond cleavage (top left). Nucleophilic attack by E104 then promotes cleavage resulting in the formation of a covalent glycosyl-enzyme intermediate (right). E55 promotes water activation and cleavage of the glycosyl-ester bond. H56 is repositioned to facilitate this second step by enforcing an interaction between E104 and the ribose hydroxyl (bottom centre). Departure of the cleaved dRP allows enzyme reset and fresh substrate binding (bottom left).

GTCAAGGATCAGCCTCGAAGTATCGATC-3'). The truncated genes were subsequently inserted into the pHAT4 expression vector[27], a gift from Marko Hyvönen & Johan Peränen.

## Protein purification

DNPH1$^{WT}$ and point mutants were expressed in *E. coli* BL21 Star™ (DE3) cells (Invitrogen), cultured in Terrific Broth (TB) supplemented with 100 μg/mL ampicillin and 0.02% anti-foam Y-30 emulsion (Sigma). Cultures (1 L) were incubated in 2 L baffled flasks at 37 °C with shaking, with growth monitored by optical density at 595 nm (OD$_{595}$). Expression was induced at OD$_{595}$ ≈ 1–1.2 using 0.5 mM Isopropyl β-ᴅ-1-thiogalactopyranoside (IPTG). Approximately 4 hrs post-induction, cell pellets were harvested by centrifugation prior to snap-freezing in liquid nitrogen and storage at −80 °C.

To purify DNPH1, cell pellets were thawed on ice and resuspended in 30 mL lysis buffer (50 mM HEPES-NaOH pH 7.5, 5% v/v glycerol, 300 mM NaCl, 0.5 mM TCEP, 20 mM imidazole), supplemented with protease inhibitor cocktail (cOmplete™, EDTA-free, Roche). Cells were lysed on ice by sonication, and the resulting lysate was clarified by centrifugation (45,000 × *g*, 1 h, 4 °C).

Initial capture from the clarified lysate was performed using batch affinity chromatography with 1.5 mL Ni-NTA agarose resin (Qiagen). Following incubation for 1 h at 4 °C, the resin was washed 5× with 12 mL lysis buffer, before the target protein was eluted using 5 × 3 mL lysis buffer supplemented with 180 mM imidazole. Protein purity was analysed by SDS-PAGE.

Pooled elution fractions (~12 mL) were subjected to affinity tag cleavage to facilitate crystallisation, using 0.5–1 mg TEV protease. This was performed overnight at 4 °C while simultaneously dialysing against ~800 mL low salt buffer (20 mM HEPES-NaOH pH 7.5, 100 mM NaCl, 5% v/v glycerol, 0.5 mM TCEP), using a 12 mL 10 kDa MWCO Slide-A-Lyzer cassette (ThermoFisher Scientific). The buffer was changed once. The protein was then passed through a 0.22 μm PES syringe filter before being passed through a 1 mL HisTrap HP column (Cytiva) and subsequently a 1 mL Resource Q anion exchange column (GE Healthcare/Cytiva) using an ÄKTA Pure FPLC (GE Healthcare/Cytiva) at 4 °C, using UNICORN software (v 7.3, Cytiva). Analysis by SDS-PAGE revealed that DNPH1 failed to bind to either column after tag removal, resulting in purified enzyme in the final flow-through which was concentrated to >8 mg/mL using a 20 mL 10 kDa MWCO PES Vivaspin device (Sartorius). Aliquots of ~10 mg protein were snap frozen in liquid nitrogen and stored at −80 °C.

Prior to use, a ~10 mg aliquot of DNPH1 was thawed and subjected to size exclusion chromatography using a HiLoad 16/600 Superdex S75 pg column (GE Healthcare/Cytiva), pre-equilibrated in gel filtration buffer (20 mM HEPES-NaOH pH 7.5, 150 mM NaCl, 0.5 mM TCEP). Fractions corresponding to the single major elution peak were analysed by SDS-PAGE, pooled and concentrated to >12 mg/mL using a 4 mL 10 kDa MWCO Amicon Ultra device (Merck). Concentrated protein was snap-frozen in 25–50 μL aliquots before storage at −80 °C.

Preliminary crystallographic studies with DNPH1$^{E55A}$ by mass spectrometry indicated the presence of contaminating nucleotides, so an additional dialysis step was performed prior to size exclusion. Dialysis was carried out overnight at 4 °C against ~45 mL 10× PBS supplemented with 0.5 mM TCEP, using a 2 mL, 10 kDa MWCO Slide-A-Lyzer MINI device (ThermoFisher Scientific). The buffer was changed 2×. This approach was utilised for all DNPH1$^{E55Q}$ samples used in this study.

## Crystallization

All vapour diffusion crystallization experiments were performed using 96-well, 2-drop 'MRC' plates. For each protein, crystallization screening trials were performed with the JCSG Plus, PACT Premier and Morpheus screens (Molecular Dimensions), using 0.1 μL crystallization buffer mixed with 0.1 μL protein (7.5 mg/mL), both with and without

2–5 mM hmdUMP (Jena Bioscience or Biosynth Carbosynth), in separate drop positions. Where necessary, optimisation screens based on hits in the presence of substrate were subsequently generated using a Formulatrix Formulator robot and dispensed as before except using protein at 5 and 7.5 mg/mL, both in the presence of 5 mM substrate. All trays were dispensed at room temperature using an NT8 humidity-controlled robot (Formulatrix) but stored at 4 °C and automatically imaged by a Formulatrix Rock Imager.

Crystal harvesting was performed at 4 °C, with crystals cryo-protected stepwise using varying ratios of crystallization buffer:cryo-protectant (2:1, 1:2, 0:1). See Supplementary Table 3 for optimised crystallization conditions and respective cryoprotectants.

## Data processing, model building and refinement, model visualisation

X-ray diffraction data for DNPH1$^{E104Q}$ were collected at 0.9763 Å and 100 K at Diamond Light Source (DLS; Didcot, Oxfordshire, UK) beamline IO3, and were processed using AutoPROC[28–32]. Data for DNPH1$^{E55Q}$ were collected at 0.8856 Å and 100 °K at DLS beamline IO4, and were processed using DUI/DIALS[33] and POINTLESS/AIMLESS[30, 31] as part of the CCP4/CCP4i2 package[32, 34]. In both cases, General Data Acquisition ('GDA', OpenGDA) software was used for data collection. Resolution cut-offs were determined automatically for the DNPH1$^{E104Q}$ dataset. Diffraction data for DNPH1$^{E55Q}$ displayed moderate levels of anisotropy, as indicated by the STARANISO server (Global Phasing Ltd). Use of anisotropic resolution cut-offs did not improve subsequent refinement, so a high-resolution spherical cut-off, as determined by the server, was used. This had some effects on data statistics, but prevented the loss of weaker data that may be of use in future refinement as software packages evolve.

Molecular replacement was performed using Phaser[35] within the Phenix software package[36]. Iterative rounds of refinement and manual model building were performed using phenix.refine[37] and Coot[38], respectively. Where necessary, ligand restraints were generated using eLBOW[39]. Validation was performed using MolProbity[40]. Model visualisation was performed using ChimeraX[41].

## Substrate turnover assays

DNPH1 substrate turnover rates were determined using an HPLC-based system to separate and quantify substrate and liberated hmU product[8]. Protein concentrations and incubation times were calibrated to account for variation in the activity levels of each mutant protein. Chromnav software (v 1.19, Jasco) was used for data collection and processing.

To determine the influence of substrate concentration on catalytic activity, a continuous UV-spectrophotometric method was used, taking advantage of the UV-absorbance shift observed when hmU is liberated from hmdUMP substrate. Triplicate reactions were performed at room temperature in a final volume of 150 uL, containing 20 mM sodium phosphate pH 7.0, 150 mM NaCl, 2 mM MgCl$_2$, 1–200 μM hmdUMP and either 100 nM (DNPH1$^{WT}$) or 1 μM (DNPH1$^{H56A}$) protein. Hydrolysis of hmdUMP was monitored by the reduction in absorbance at 275 nm in a 1 cm path length quartz cell using a Jasco V-760 spectrophotometer recording at 1 s intervals. Reactions were monitored for a minimum of 5 min to ensure data quality. The turnover rate was calculated from the first ~30 secs of each data set to minimise the effects of depletion at low substrate concentrations. SpectraManager software (v 2,5, Jasco) was used for data collection and initial processing. Non-linear regression was used to fit the standard Michaelis-Menten model to the data using GraphPad Prism (GraphPad Software, San Diego).

## Simplified mechanistic model

To explore the roles played by individual residues in substrate turnover, we define a simplified mechanistic pathway to identify the key

rate-limiting steps (Eq. 1):

$$E + S \underset{k_{-1}}{\overset{k_1}{\rightleftharpoons}} ES \overset{k_2}{\longrightarrow} EI + P_1 \overset{k_3}{\longrightarrow} E + P_2 \tag{1}$$

In this model, the reversible interaction of substrate (S) and enzyme (E) to yield the Michaelis complex (ES) can be described in terms of the rate constants of association ($k_1$) and dissociation ($k_{-1}$). The ratio $k_{-1} : k_1$ determines the affinity ($K_D$) of enzyme for substrate. The first catalytic step yields the glycosyl-enzyme intermediate (EI), the formation of which is determined by the rate constant $k_2$. We assume this step is irreversible, and for simplicity do not consider the release of the cleaved base ($P_1$). The second irreversible catalytic step cleaves the sugar-phosphate ($P_2$) from the enzyme and is defined by rate constant $k_3$. We do not separately consider subsequent product release as this is assumed to be faster than cleavage of the intermediate.

### Stopped-flow analyses

Stopped-flow experiments were conducted on a HiTech SF61 DX2 apparatus, equipped with a mercury-xenon lamp (TgK Scientific Ltd). DNPH1 (DNPH1[WT], DNPH1[E104Q], DNPH1[H56A] or DNPH1[E55Q]) solutions were rapidly mixed with different concentrations of hmdUMP in the stopped-flow instrument and intrinsic tryptophan fluorescence was recorded over short (2 or 5 s) and long (50 s) timescales. Tryptophan fluorescence was excited at 295 nm (1 mm slits) and emission light was detected with two photomultipliers simultaneously, after filtering scattered light through 320 nm longpass filters (Schott); signals from the two channels were averaged. A minimum of three fluorescence traces were acquired for each substrate concentration and time range, and were averaged before analysis. Data acquired on a 50-s timescale were corrected for photobleaching by subtracting the fluorescence traces measured in the absence of substrate.

Stopped-flow traces were analysed by non-linear least square fitting in Kinetic Studio 5.0 (TgK Scientific Ltd) to determine observed rate constants. The fluorescence decreases seen for DNPH1[E104Q] and DNPH1[H56A] were analysed using a single exponential plus a linear function to yield an observed rate constant for substrate binding. The linear component accounts for a small percentage of photobleaching during the short timescale. For DNPH1[WT] and DNPH1[E55Q], a double exponential function was used for fitting the biphasic data. In the case of DNPH1[WT] the two rate constants from the fit were used as estimates of the observed rate constants of substrate binding and substrate turnover. For DNPH1[E55Q], the observed rate constant of the second step is slower and was estimated by fitting a single exponential plus a linear function to the data on a long (50 s) timescale. Secondary plots of observed rate constants versus substrate concentrations and linear regressions were performed in GraphPad Prism. Representative fluorescence traces were also plotted in Prism after normalising the fluorescence values to the Y-intercept of the exponential fit.

All measurements were performed at least in triplicate at 25 °C in buffer consisting of 20 mM sodium phosphate pH 7.0, 150 mM NaCl and 2 mM MgCl$_2$. Concentrations quoted in the text and figures refer to the final (mixing chamber) concentrations.

### Mass spectrometry

The presence (and reduction) of contaminating AMP and/or GMP in DNPH1 was assayed by LC-MS/MS[42]. Samples were injected into a Dionex UltiMate LC system (Thermo Scientific) using a ZIC-pHILIC (150 mm × 4.6 mm, 5 μm particle) column (Merck Sequant). A 15 min elution gradient was used (80% solvent A to 20% solvent B), followed by a 5 min wash (95:5 solvent A to solvent B) and 5 min re-equilibration. Solvent A was 20 mM ammonium carbonate in water (Optima HPLC grade, Sigma Aldrich) and solvent B was acetonitrile (Optima HPLC grade, Sigma Aldrich). Flow rate was 300 μl/min; column temperature

was 25 °C; injection volume was 10 μl; and autosampler temperature was 4 °C. The LC system was coupled to a TSQ Quantiva Triple Quadrupole (Thermo Scientific) mass spectrometer. The heated electrospray ionisation (HESI) interface was operated in positive mode with a spray voltage of 3500 V, a capillary temperature of 375 °C, a vaporiser temperature of 275 °C, auxiliary gas pressure (arbitrary units) of 45, sheath gas pressure (arbitrary units) of 16, ion sweep gas pressure (arbitrary units) of 5 and CID gas set at 1.5 mTorr. The MS/MS was operated in Selected Reaction Monitoring (SRM) mode with the transitions $m/z$ 348.1 (AMP) → $m/z$ 136.1 and $m/z$ 364.1 (GMP) → $m/z$ 152.1, both with collision energy 18 V and dwell time 40 ms. Xcalibur Qual Browser and Tracefinder 4.1 software (Thermo Scientific) were used for qualitative and quantitative analysis respectively, according to the manufacturer's workflows.

Calibration samples were prepared by spiking samples diluted 50 times with AMP or GMP solutions to concentrations of 0.0032, 0.016, 0.08, 0.4, 2 and 10 μM (in addition to endogenous level). Linearity was evaluated and the endogenous concentrations were determined at the x-intercept of the response curve generated.

### Bioinformatics

Alignment of DNPH1 sequences was performed using Clustal Omega[43] and subsequent visualisation was performed using Jalview. Structural topology analysis was performed using PDBsum[44]. 3D-conservation analysis was performed using ConSurf[45], using a default run against the SwissProt database. Ligand interaction analyses were performed using LigPlot+[46].

### Reporting summary

Further information on research design is available in the Nature Portfolio Reporting Summary linked to this article.

## Data availability

Atomic coordinates and structure factors generated in this study for DNPH1[E104Q] and DNPH1[E55Q] have been deposited in the Protein Data Bank (PDB) under accession codes 8QHQ [https://doi.org/10.2210/pdb8qhq/pdb] and 8QHR [https://doi.org/10.2210/pdb8qhr/pdb] respectively. The processed enzyme activity data (HPLC data, UV-spectrophotometric data and stopped-flow data) are provided in the Source Data file. Source data are provided with this paper.

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

## Acknowledgements

The authors would like to thank Diamond Light Source for beamtime (proposal mx25587), and the staff of beamlines I03, I04 and I24 for assistance with crystal testing and data collection. We thank Phil Walker and the Structural Biology STP for their support throughout the study. Work in S.C.W.'s laboratory was supported by the Francis Crick Institute which receives core funding from Cancer Research UK (CC2098), the UK Medical Research Council (CC2098) and the Wellcome Trust (CC2098); by the European Research Council (ERC-ADG-666400); and by the Louis-Jeantet Foundation. For the purpose of Open Access, the authors have applied a CC BY public copyright licence to any Author Accepted Manuscript version arising from this submission.

## Author contributions

N.J.R. and S.C.W. designed the study. N.J.R. purified proteins and carried out biochemical and biophysical analyses, with contributions from S.K., and I.A.T. N.J.R. performed X-ray crystallography experiments and data analyses, with contributions from A.P. N.J.R. solved and refined the

crystal structures. M.S.D.S performed MS experiments and analysed the data, with contributions from J.M. S.C.W. supervised the experiments and data analyses. N.J.R. and S.C.W. prepared the manuscript with contributions from all authors.

## Funding

## Competing interests
K.F. and S.C.W. are inventors on patent WO2021/048235 that pertains to the use of DNPH1 inhibitors and hmdU as a mechanism to sensitise HR-deficient cells to PARPi. The remaining authors declare no competing interests.
