## [Peer Review File · Nature Communications]

Mechanism of substrate hydrolysis by the human nucleotide pool sanitiser DNPH1REVIEWERS' COMMENTS

Reviewer #1 (Remarks to the Author):

The authors have recently identified that human nucleotide glycosidase DNPH1 as an important drug target for treatment of BRCA-deficient cancers by potentiating the PARPi activity. In this study they provided two sets of crystallographic structures of DNPH1 mutants complexed with the substrate at two key stages of the catalytic cycle. The structure of the enzyme-deoxysugar monophosphate intermediate is clearly snapped. Based on comparative analysis of the structures of MilB, authors determined that the second catalytic cycle is rely on the E55 residue other than a D, that is also a D at the first catalytic cycle for the classic family of N-glycosidase. In combination with kinetic assays on the single-point mutants, authors demonstrated the whole catalytic cycle of this enzyme, wherein the role of these key sites are depicted and discussed, these mechanistic insights will facilitate rational drug design to specifically inhibit the activity of DNHP1. Therefore, this important study deserves a rapid publication in Nature Communications.

Minor point:

1, DNPH1 targets HmdUMP at a monophosphate level while incorporation of hmdU into DNA should occur at a level of triphosphates. Authors need to detail the possible pathway how deletion of DNPH1 leads to incorporation of hmdU into DNA.。

2, There are two tandem 'biologically' in one sentence in page 8.

Reviewer #2 (Remarks to the Author):

Inhibition of poly(adenosine diphosphate–ribose) polymerase (PARP) for treatment of tumors with compromised DNA repair by the homologous recombination pathway is used in clinic. The authors have previously identified DNPH1 as an enzyme that causes resistance to PARP inhibitors leading them to propose that DNPH1 inhibitors could potentiate the PARP inhibitors and would therefore be of interest in the treatment of ovarian breast and prostate cancers.

In the manuscript entitled: "Mechanism of substrate hydrolysis by the human nucleotide pool sanitiser DNPH1" the authors solved the structure of the enzyme in complex with its substrate ie 5-hydroxymethyl-deoxyuridine 5' monophosphate. They note that the dimer has a similar fold as reported previously by others and that the substrate binding is close to that of MilB a *Streptomyces rimofaciens* enzyme that hydrolyzes 5-hydroxymethyl-cytidine 5' monophosphate and to a lesser extent 5-hydroxymethyl-deoxycytidine 5' monophosphate. Based on sequence comparison, the authors identified two conserved aminoacids R30 and H56 which, when mutated, alter the catalytic rate. They next show that the Histidine 56 is involved in the catalysis and propose that it may stabilize the hmU leaving group during glycosidic bond cleavage. A more detailed analysis of the region surrounding the aminoacid H56 led them to investigate the role of the glutamate 55 which is conserved in metazoan DNPH1. Interestingly, they show that both E55 and H56 are involved in the catalysis but at different stages: H56 being important in the first catalytic step ie the glycosidic bond cleavage while E55 is important in the second ie the glycosyl-enzyme intermediate cleavage. In the last part, the authors solved the structure of DNPH1 E55Q co-crystallised with hmdUMP using the E104Q as a model for molecular replacement. Based on the glycosyl-enzyme intermediate trapped, the authors propose E55 as a novel contributor of the active site altering the understanding of the catalytic mechanism of the enzyme.

This mechanism differs from that described for this family of enzymes such as nucleoside 2'-deoxyribosyltransferases (NTDs) and from that published by Devi et al. (Biochemistry 2023, 62, 17, 2658-2668) on the same human DNPH1 involving Y24, D80 and E104.

As mentioned by the authors this model is important for the future development of DNPH1 inhibitors by rational design. Unfortunately, no molecules have been synthesized and tested to date, which

would have added value to the manuscript, but this will certainly be the subject of future research. The manuscript is very clear, the experiments are well conducted, the data are robust and well presented and the conclusions are appropriate. However, one wonders whether it would be better suited to a more specialized journal, but that is the editor's decision.

We sincerely thank the reviewers for their careful reading of the manuscript. It is very rewarding to see their appreciation that the paper represents a significant advance, and of course we are delighted to see their enthusiastic support for publication in *Nature Comms*.

The Response to Reviewers is as follows (reviewer comments in black, and our response is in blue):

Referee #1:

The authors have recently identified that human nucleotide glycosidase DNPH1 as an important drug target for treatment of BRCA-deficient cancers by potentiating the PARPi activity. In this study they provided two sets of crystallographic structures of DNPH1 mutants complexed with the substrate at two key stages of the catalytic cycle. The structure of the enzyme-deoxysugar monophosphate intermediate is clearly snapped. Based on comparative analysis of the structures of MilB, authors determined that the second catalytic cycle is rely on the E55 residue other than a D, that is also a D at the first catalytic cycle for the classic family of N-glycosidase. In combination with kinetic assays on the single-point mutants, authors demonstrated the whole catalytic cycle of this enzyme, wherein the role of these key sites are depicted and discussed, these mechanistic insights will facilitate rational drug design to specifically inhibit the activity of DNHP1. Therefore, this important study deserves a rapid publication in Nature Communications.

Thank you for the kind words and support.

Minor point:
1, DNPH1 targets HmdUMP at a monophosphate level while incorporation of hmdU into DNA should occur at a level of triphosphates. Authors need to detail the possible pathway how deletion of DNPH1 leads to incorporation of hmdU into DNA.

We now note on p2 the following 'The cellular role of DNPH1 is to remove hydroxymethyl deoxyuridine monophosphate (hmdUMP) from the nucleotide pool. This activity prevents a cascade of nucleotide phosphorylation events that involves deoxythymidylate kinase (DTYMK), thereby limiting the incorporation of hmdU into genomic DNA⁸.

2, There are two tandem 'biologically' in one sentence in page 8.

The typo has been corrected.

Referee #2:

Inhibition of poly(adenosine diphosphate–ribose) polymerase (PARP) for treatment of tumors with compromised DNA repair by the homologous recombination pathway is used in clinic. The authors have previously identified DNPH1 as an enzyme that causes resistance to PARP inhibitors leading them to propose that DNPH1 inhibitors could potentiate the PARP inhibitors and would therefore be of interest in the treatment of ovarian breast and prostate cancers. In the manuscript entitled: "Mechanism of substrate hydrolysis by the human nucleotide pool sanitiser DNPH1" the authors solved the structure of the enzyme in complex with its substrate ie 5-hydroxymethyl-deoxyuridine 5' monophosphate. They note that the dimer has a similar fold as reported previously by others and that the substrate binding is close to that of MilB a *Streptomyces rimofaciens* enzyme that hydrolyzes 5-hydroxymethyl-cytidine 5' monophosphate and to a lesser extent 5-hydroxymethyl-deoxycytidine 5' monophosphate. Based on sequence comparison, the authors identified two conserved aminoacids R30 and H56 which, when mutated, alter the catalytic rate. They next show that the Histidine 56 is involved in the catalysis and propose that it may stabilize the hmU leaving group during glycosidic bond cleavage. A more detailed analysis of the region surrounding the aminoacid H56 led them to investigate the role of the glutamate 55 which is conserved in metazoan

DNPH1. Interestingly, they show that both E55 and H56 are involved in the catalysis but at different stages: H56 being important in the first catalytic step ie the glycosidic bond cleavage while E55 is important in the second ie the glycosyl-enzyme intermediate cleavage. In the last part, the authors solved the structure of DNPH1 E55Q co-crystallised with hmdUMP using the E104Q as a model for molecular replacement. Based on the glycosyl-enzyme intermediate trapped, the authors propose E55 as a novel contributor of the active site altering the understanding of the catalytic mechanism of the enzyme.

This mechanism differs from that described for this family of enzymes such as nucleoside 2'-deoxyribosyltransferases (NTDs) and from that published by Devi et al. (Biochemistry 2023, 62, 17, 2658-2668) on the same human DNPH1 involving Y24, D80 and E104. As mentioned by the authors this model is important for the future development of DNPH1 inhibitors by rational design. Unfortunately, no molecules have been synthesized and tested to date, which would have added value to the manuscript, but this will certainly be the subject of future research.

The manuscript is very clear, the experiments are well conducted, the data are robust and well presented and the conclusions are appropriate.

However, one wonders whether it would be better suited to a more specialized journal, but that is the editor's decision.

Thank you for the kind words. The structures presented in this paper are already being used, both in my laboratory and at AstraZeneca, in attempts to identify and develop new small molecule inhibitors of DNPH1.